# Intestinal Barrier Impairment Induced by Gut Microbiome and Its Metabolites in School-Age Children with Zinc Deficiency

**DOI:** 10.3390/nu16091289

**Published:** 2024-04-26

**Authors:** Xiaoqi Chai, Xiaohui Chen, Tenglong Yan, Qian Zhao, Binshuo Hu, Zhongquan Jiang, Wei Guo, Ying Zhang

**Affiliations:** 1School of Public Health, Lanzhou University, Lanzhou 730000, China; chaixq21@lzu.edu.cn (X.C.); chenxiaohuiiii@163.com (X.C.); yantlyan@163.com (T.Y.); zhaoq2512@163.com (Q.Z.); hubsh1997@163.com (B.H.); yxxwhgz@outlook.com (Z.J.); 2Key Laboratory of Animal Genetics, Breeding and Reproduction in the Plateau Mountainous Region, Ministry of Education, Guizhou University, Guiyang 550000, China

**Keywords:** intestinal microbiota, metabolites, zinc deficiency, school-age children

## Abstract

Zinc deficiency affects the physical and intellectual development of school-age children, while studies on the effects on intestinal microbes and metabolites in school-age children have not been reported. School-age children were enrolled to conduct anthropometric measurements and serum zinc and serum inflammatory factors detection, and children were divided into a zinc deficiency group (ZD) and control group (CK) based on the results of serum zinc. Stool samples were collected to conduct metagenome, metabolome, and diversity analysis, and species composition analysis, functional annotation, and correlation analysis were conducted to further explore the function and composition of the gut flora and metabolites of children with zinc deficiency. Beta-diversity analysis revealed a significantly different gut microbial community composition between ZD and CK groups. For instance, the relative abundances of *Phocaeicola vulgatus*, *Alistipes putredinis*, *Bacteroides uniformis*, *Phocaeicola* sp000434735, and *Coprococcus eutactus* were more enriched in the ZD group, while probiotic bacteria *Bifidobacterium kashiwanohense* showed the reverse trend. The functional profile of intestinal flora was also under the influence of zinc deficiency, as reflected by higher levels of various glycoside hydrolases in the ZD group. In addition, saccharin, the pro-inflammatory metabolites, and taurocholic acid, the potential factor inducing intestinal leakage, were higher in the ZD group. In conclusion, zinc deficiency may disturb the gut microbiome community and metabolic function profile of school-age children, potentially affecting human health.

## 1. Introduction

As a vital micronutrient for living organisms, zinc has powerful biological functions. In eukaryotes, about 10% of protein is zinc-dependent and most of it belongs to enzymes and transcription factors, which suggests the significance of zinc in cellular functions [1,2]. In addition, zinc plays important roles in antioxidative processes [3], cell growth and proliferation [4], and intracellular signaling [5]. However, at least one billion people worldwide suffered from zinc deficiency [6] and it was reported that 38.2% of Chinese children were zinc-deficient [7]. School age is a critical time for children’s physical growth and mental development [8], and school-age children with zinc deficiency are at high risk of parasite infection, diarrhea, and pneumonia [9,10]. Zinc deficiency is associated with serious health consequences, which is the main driver for child mortality in developing countries [11]. Therefore, addressing zinc deficiency in children is more beneficial to future generations.

As a research hotspot in recent years, intestinal flora was a crucial factor in the development of diseases [12], providing a new strategy for exploring nutritional problems such as zinc deficiency. Low levels of dietary zinc have been reported to alter the microbial composition of the gut, specific taxonomic abundance, and overall microbial diversity in mice and chicks [13,14,15]. Furthermore, zinc deficiency may cause a change in microbial function via altered gut microbiota [16]. Functional predictive analysis showed that certain Kyoto Encyclopedia of Genes and Genomes (KEGG) metabolic pathways were depleted among zinc-deficient mice, such as lipid metabolism, mineral absorption, and bile acid biosynthesis [15]. In addition, the levels of acetate and hexanoate were significantly decreased in the cecum of zinc-deficient mice. A short-term low-Zn diet resulted in the decrease in pathways related to carbohydrate, glycan, and nucleotide metabolism [13].

Besides the changes in intestinal microbes and metabolites described above, zinc is inextricably linked with the integrity of the gut barrier. Recent research has demonstrated that zinc deficiency may influence intestinal barrier function via altering gastro-intestinal (GI) markers in vivo and vitro models [17,18]. On the contrary, zinc supplementation in patients with Crohn’s disease could improve intestinal permeability [19] and zinc may reverse gut barrier injury through increased levels of tight junction proteins such as OCLD in duck models [20]. Therefore, zinc deficiency may affect gut health by disrupting the intestinal barrier.

In a previous study [21], we already found that zinc deficiency may cause alterations in the intestinal microbiota and function in school-age children. However, whether these dynamic variations affected gut microbiome metabolites and even the intestinal barrier is still unclear. Thus, to further investigate the effects of zinc deficiency on gut microbes and bacterial metabolites, and to determine whether it is associated with intestinal barrier injury, this study screened participants from a previous study and conducted metagenomic and metabolomics analysis to identify the characteristics of intestinal microbiota and its metabolome in school-age children with zinc deficiency.

## 2. Materials and Methods

### 2.1. Study Population and Sampling

In our previous study [21], we found that zinc deficiency leads to alterations in gut microbial diversity and composition. To further clarify the effects of zinc deficiency on the structure and function of the gut flora and its metabolites, we selected 15 extremely zinc-deficient school-age children as the zinc-deficient group based on the results of serum zinc levels and 16S rDNA sequencing in 67 school-age children, and matched them with 15 school-age children with normal zinc levels as the control group for further analysis (Appendix A). Height and weight were measured to calculate HAZ, BMI, WAZ, and BMIZ. Blood samples were taken while the participant was fasting, after centrifugation at 3000× *g* for 5 min, and serum was obtained. The Medical Ethics Committee, Lanzhou University, agreed to all protocols (GW-20200915-1), and the children’s legal guardians provided informed consent. The registration number (ChiCTR2200056909) was obtained from the Chinese Clinical Trial Registry (http://www.chictr.org.cn/index.html) (accessed on 1 May 2022). The detailed process of population recruitment, sample collection, and serum tests was previously reported [21].

### 2.2. Serum Test

Enzyme-linked immunosorbent assay (ELISA) kits (Elabscience, Wuhan, China) were used to detect inflammatory cytokines and zinc content was determined by means of a flame atomic absorption spectrometer. Serum zinc concentration cut-off values have been reported [22] as follows: 65 μg/dL for children under 10 years of age, and 70 μg/dL for females and 74 μg/dL for males over 10 years old. Stool was sampled to perform metagenomic and metabolomics analysis.

### 2.3. DNA Extraction, Metagenome Sequencing, and Data Processing

DNA was extracted from 300 mg of fecal samples. Total genomic DNA was extracted using the QIAamp DNA Stool Mini Kit (QIAGEN, Hilden, Germany), following the manufacturer’s instructions. The concentration and purity of genomic DNA (concentration ≥ 25/total amount ≥ 2, OD260/280 = 1.7~1.9) were tested using NanoDrop 2000 (Thermo Fisher Scientific, Waltham, MA, USA) and Qubit3.0 Fluorometer (Thermo Fisher Scientific, Waltham, MA, USA). To determine genomic DNA integrity, 1% agarose gel electrophoresis was used. Covaris ME220 (Covaris, Woburn, MA, USA) was used to fragment the genomic DNA into segments of approximately 400 bp in length. The metagenomic library was constructed from the NEBNext Ultra DNA Library Prep Kit (NEB, Ipswich, MA, USA).The concentration (>5 ng/μL) and fragment length (300–500 bp) of the library were determined using the Invitrogen Qubit spectrophotometer (Thermo Fisher Scientific, Waltham, MA, USA) and Agilent 2100 bioanalyzer (Agilent Technologies, Santa Clara, CA, USA). Five samples were discarded in both the ZD group and CK group because of failed DNA extraction and unqualified quality. The sequencing was conducted at Genesky Biotechnologies Inc. (Shanghai, China) on the Illumina Hiseq platform (2 × 150 bp).

The quality of the raw sequencing data was assessed by means of FastQC software (V.0.11.8, Babraham Institute, Cambridge, UK) [23]. Seqtk (V.1.3-r106, Wei Shen, Chongqing, China) [24] was used to calculate the quality score of bases and the sequencing error rate. FASTQ (V.0.19.5, HaploX, Shenzhen, China) [25] was used to remove low-quality bases (quality score < 20), short reads (<75 bp), and “N” records. The clean reads were assembled into contigs for each sample by metaSPAdes (V.3.15.1, Center for Algorithmic Biotechnology, Petersburg, Russia) [26]. From the assembled contigs >100 bp, open reading frames (ORFs) were predicted using MetaGeneMark (V.3.38, GENE PROBE Inc., Atlanta, GA, USA) [27]. All predicted genes set with a 95% sequence identity (90% coverage) were bunched into the non-redundant gene catalog according to CD-HIT (V.4.8.1, http://www.bioinformatics.org/cd-hit/, accessed on 15 July 2021) [28]. Clean reads were mapped to non-redundant genes to estimate the abundances using Salmon (V.1.2.1, https://salmon.readthedocs.io/en/latest/salmon.html, accessed on 15 July 2021) [29].

### 2.4. Taxonomic and Functional Annotation, and Bioinformatics Analysis

Functions were annotated with DIMOND (V.0.9.36, http://ab.inf.unituebingen.de/software/diamond, accessed on 20 July 2021) against the Carbohydrate-Acting Enzyme (http://wwwcazy.org/, accessed on 20 July 2021) [30] and KEGG (http://www.genome.jp/kegg/, accessed on 20 July 2021) [31] databases with an E-value ≤ 1 × 10^−5^. The abundances of KEGG Orthology (KO), KEGG pathways, and enzymes were normalized to counts per million reads (cpm) for downstream analysis. KO, KEGG pathways, and enzymes with cpm > 5 in at least 50% of children in every group were included in downstream analysis.

Taxonomic assessment of the metagenomic samples was performed using Kraken2 [32] against the GTDB (https://gtdb.ecogenomic.org/, accessed on 25 July 2021) and RefSeq database (http://www.ncbi.nlm.nih.gov/RefSeq/, accessed on 25 July 2021). The reads were annotated into bacteria, archaea, fungi, and viruses, respectively, and taxonomic profiles were performed on the level of Phylum, Class, Order, Family, Genus, and Species, with calculation of relative abundance. Microbial taxa with a relative abundance >0.1% in at least 50% of children in both groups were included in the downstream analysis.

The R software (Vegan package, V.4.0.5, R Core Team, Vienna, Austria) [33] was used to calculate the Coverage index, Shannon index, and Simpson index of alpha. The beta diversity of microbial communities was assessed using Principal Coordinate Analysis (PCoA) on the basis of the Bray–Curtis distance. Differential analysis of gut microbiota, KO, KEGG pathways, and enzymes between ZD and CK groups was performed using R software (DESeq package, V.4.0.5, R Core Team, Vienna, Austria) [34]. Circos plots of functional abundance were drawn by the circos (http://mkweb.bcgsc.ca/tableviewer/, accessed on 27 July 2021) online tool [33].

### 2.5. Fecal Metabolomics and Analysis

After being thawed, 50 mg of stool was weighted on an EP tube, and 800 μL of 80% methanol was poured in. These were followed by homogenization at 70 Hz for 90 s, followed by sonication at 4 °C for 30 min. After being hatched for 1 h at −40 °C, the samples were vortexed for 30 s and incubated for half an hour at 4 °C. Centrifugation was then performed at 12,000× *g* for 15 min at 4 °C. After adding 5 μL of internal standard (0.14 mg/mL DL-o-Chlorophenylalanine), the supernatant (200 μL) was placed in a new glass vial for liquid chromatography–mass spectrometry (LC-MS) analysis. To prepare the quality control (QC) sample, an equal aliquot of the supernatant from each sample was mixed.

LC-MS analyses were conducted on an UPLC system fitted to an ACQUITY UPLC HSS T3 column (2.1 mm × 100 mm, 1.8 μm) coupled to an Orbitrap mass spectrometer. In addition, 0.05% formic acid in water (A) and acetonitrile (B) were used as the mobile phase. The temperature of the autosampler was set to 4 °C and the volume of the injected sample was 2 μL. The flow was set to 0.3 mL/min. The gradient was set to 5% B in 0–1 min, 5–95% B in 1–12 min, 95% B in 12–13.5 min, 95–5% B in 13.5–13.6 min, and 5% B in 13.6–16 min. Both positive modes and negative modes of an electrospray ionization (ESI) source were operated. The ESI source conditions were set to the following: sheath gas flow rate, 45 Arb; aux gas flow rate, 15 Arb; capillary temperature, 350 °C; heater temperature, 300 °C; sweep gas flow rate, 1 Arb; and spray voltage, 3.0 kV (positive) or −3.2 kV (negative). The scan range was 70–1050 m/z. The fecal metabolites were measured at Genesky Biotechnologies Inc. (Shanghai, China).

All single peaks were filtered and the peak area data were discarded where a single group’s null value exceeded 50%, or where all groups’ null values exceeded 50%. Missing values in the original data were filled in using the minimum two-thirds method, and then the data were standardized using the sum normalization method. Then, the metabolites were annotated with reference to the HMDB (Human Metabolome Database) [35] and the KEGG database. Finally, multidimensional statistical analysis was conducted on two groups of metabolites by PLS-DA. The difference in metabolites between the group with zinc deficiency and the control group was analyzed using R software [34]. A corrected *p*-value of less than 0.05 was considered a significant difference between the two groups.

### 2.6. Statistical Analyses

Questionnaire double entry was performed using EpiData 3.1 software. NARs, MAR, diversity indices, and serum indicators are expressed in terms of the mean and standard error of mean (SEM), and statistical analyses were conducted using Student’s *t* test. Z-scores are expressed as medians, and statistical analyses were performed using the Mann–Whitney U test. All statistical analyses above were conducted using GraphPad Prism 8.0 and SPSS 24.0, and a difference of *p* < 0.05 was regarded as statistically significant.

## 3. Results

### 3.1. Characteristics of Study Populations

From 67 children reported in the previous study [21], 30 school-age children were subjected to gut metagenome and metabolome analyses. The demographic characteristics of participants are shown in Table 1. The ZD group included 8 females and 7 males, with a mean age of 8.20 ± 0.28 years. In contrast, the CK group comprised 7 females and 8 males, with a mean age of 9.07 ± 0.40 years. For Z-score, only the BMIZ score was significantly different in the two groups. In the ZD group, serum zinc was 22.94 ± 4.67 μg/dL, significantly lower than that of the CK group (*p* < 0.001), while the level of TNF-α was significantly higher than that of the CK group (*p* < 0.001).

### 3.2. Profiling of Gut Metagenome in ZD and CK Group

A total of 165,939,340.8 reads were generated by metagenomic sequencing. This corresponds to 829,696.70 ± 348,899.7 reads (mean ± SEM) per sample. After quality control and the removal of host genes, a total of 156,149,749.2 reads were retained, and 780,748.75 ± 355,480.3 reads per sample were retained. A total of 808,260.7 contigs (N50 length 2417 ± 239 bp) were generated by de novo assembly, with 404,130 ± 196.95 contigs per sample. The intestinal metagenome consisted of 99.36% bacteria (1,378,293 gene count), 0.22% archaea (3029 sequences), 0.11% viruses (1509 sequences), 0.09% fungi (1262 sequences), and 0.22% others (3120 sequences).

### 3.3. Compositional Profiles of the Gut Microbiome and Taxonomic Differences between the ZD and CK Group

The alpha diversity failed to differ significantly between the two groups (Figure 1A,B), and separations between the two groups based on bacteria were found using Principal Coordinate Analysis (PCoA) (*p* < 0.001, PERMANOVA; Figure 1C). In the ZD group, the first four dominant bacterial phyla included *Firmicutes* (61.74%), *Bacteroidota* (26.05%), *Actinobacteriota* (7.50%), and *Proteobacteria* (1.55%), while *Firmicutes* (69.82%), *Actinobacteriota* (20.77%), *Bacteroidota* (5.70%), and *Proteobacteria* (1.22%) were the most dominant phyla in the CK group (Appendix A). Of these, the relative abundance of *Bacteroidota* was increased by a significant amount in the ZD group, and *Actinobacteriota* was increased by a significant amount in the CK group (*p* < 0.05).

At the species level, there were 29 differential microbial species between ZD and CK groups (Figure 1D). *Phocaeicola vulgatus*, *Alistipes putredinis*, *Bacteroides uniformis*, *Phocaeicola* sp000434735, *Coprococcus eutactus*, *KLE1615* sp900066985, *Agathobacter faecis*, *Roseburia intestinalis*, *Prevotella* sp900318395, *Gemmiger qucibialis*, *Faecalibacterium prausnitzii_D*, *CAG-177* sp003538135, *Agathobacter* sp900546625, *Faecalibacterium prausnitzii_C*, *Ruminococcus* sp900318825, and Unassigned were more likely to be found in the ZD group (*p* < 0.05). Species enriched in the CK group (*p* < 0.05), in comparison with the ZD group, belonged mainly to the probiotic bacteria *Bifidobacterium* (*Bifidobacterium kashiwanohense*, *Bifidobacterium catenulatum*, *Bifidobacterium* sp002742445, *Bifidobacterium kashiwanohense_A*, *Bifidobacterium pseudocatenulatum*), *Blautia* (*Blautia_A massiliensis*, *Blautia_A* sp900066335), *Dialister* (*Dialister* sp900543455, *Dialister invisus*), *Dorea*_*A longicatena*, *Enorma* sp900538305, *Anaerobutyricum hallii*, and *Ligilactobacillus ruminis*.

### 3.4. KEGG Functional Profiles of the Gut Microbiome and Differential Functions between the ZD and CK Group

Based on the KEGG database, functional annotation of the gut microbiome in the ZD and CK group was performed. For the KEGG level 1 pathways (Appendix A), there were Metabolism (41.95%), Environmental Information Processing (19.72%), Genetic Information Processing (15.95%), Cellular Processes (10.69%), Human Diseases (7.68%), and Organismal Systems (4.01%).

For the KEGG level 2 pathways, a total of 49 secondary metabolic pathways were annotated, and the Top 20 pathways were Protein Families: Signaling and Cellular Processes; Protein Families: Genetic Information Processing and Carbohydrate Metabolism; Protein Families: Metabolism, Membrane Transport, Signal Transduction, Amino Acid Metabolism, Metabolism of Cofactors and Vitamins, Cellular Community-Prokaryotes, Translation, Glycan Biosynthesis and Metabolism, Energy Metabolism, Nucleotide Metabolism, Replication and Repair, Lipid Metabolism, Folding, Sorting, and Degradation; and Drug Resistance: Antimicrobial, Metabolism of Other Amino Acids, Biosynthesis of Other Secondary Metabolites, and Cell Growth and Death (Appendix A). For the KEGG level 3 pathways, a total of 490 pathways were obtained, of which 134 metabolic pathways were significantly different between the ZD group and CK group (Appendix A). Figure 2A shows the enriched Top 20 metabolic pathways in both groups.

In addition, Figure 2B shows a schematic diagram of the taurine and hypotaurine metabolic pathways in the KEGG database. Among them, taurocholic acid was generated in hepatocytes through a series of biochemical reactions, and it was transported to the bile tubules through receptors on the cell membrane. On the one hand, taurocholic acid could enter the small intestine through the bile tubules, where it participated in the digestion and absorption of fat, biosynthesis of secondary bile acids, and finally excreted through feces. On the other hand, taurocholic acid was reabsorbed by bile duct cells in the lumen of the bile tubules, excreted by it, reached the kidneys, and finally excreted in urine. Figure 2C shows the related KOs in the taurine and hypotaurine metabolic pathways, among which Lanine Dehydrogenase (K00259), Phosphate Acetyltransferase (K00625), Gamma-glutamyltranspeptidase (K00681), Acetate Kinase (K00925), and Glutamate Decarboxylase (K01580) were markedly higher in the ZD group (*p* < 0.05), while the levels of Holoylglycine Hydrolase (K01442) were higher in the ZD group, though the difference was not statistically significant. In the CK group, the levels of Sodium/bile Acid Cotransporter (K14342) were markedly higher (*p* < 0.05), while the levels of Sodium/glucose Cotransporter (K14158) were higher in the control group, though the difference was not statistically significant.

### 3.5. CAZy Functional Profiles of the Gut Microbiome and Differential Functions between the ZD and CK Group

Functional profiles of gut microbes in the ZD and CK groups were analyzed based on the Carbohydrate-Active Enzyme Database (CAZy). Appendix A outlines the distribution of enzymes in the samples. The highest proportion was observed for glycosyl transferases (GTs) at 42.44%, followed by glycoside hydrolases (GHs) at 38.53%, carbohydrate-binding modules (CBMs) at 13.37%, carbohydrate esterases (CEs) at 4.19%, auxiliary activities (AAs) at 0.76%, and polysaccharide lyases (PLs) at 0.71%.

After functional annotation and screening, 244 carbohydrate-active enzymes were obtained, including 134 GHs, 46 GTs, 34 CBMs, 14 CEs, 12 PLs, and 4 AAs. The results of the differential analysis of carbohydrate-active enzymes between the two groups are presented in Figure 3. As shown, PL and GH were markedly higher in the ZD group (*p* < 0.05). AA was higher in the CK group (*p* < 0.05), and the CK group exhibited higher levels of CE, GT, and CBM, though the differences were not statistically significant.

Figure 3B,C show the levels of certain CBM and GH in the ZD and CK groups. As can be seen from the figures, the levels of carbohydrate-active enzymes such as CBM67, CBM62, CBM61, CBM6, CBM4, CBM35, CBM32, and CBM20 were markedly higher in the ZD group (*p* < 0.05). The amounts of carbohydrate-active enzymes such as GH95, GH29, GH33, GH35, GH16, GH20, GH101, GH89, GH85, and GH18 were markedly higher in the ZD group (*p* < 0.05). Among them, GH95 and GH29 belonged to Fucosidases; GH33 belonged to Sialidases; GH35, GH16, and GH20 belonged to Galactosidases; GH101 belonged to N-acetylsalicylases; and GH89, GH85, and GH18 belonged to N-acetylglucosaminidase.

### 3.6. Profile of Gut Metabolome and Differential Metabolites between the ZD and CK Groups

Metabolite levels of intestinal flora from school-age children in the ZD and CK groups were investigated by LC-MS. A total of 5675 peaks were extracted in the positive ion mode and 6189 peaks were extracted in the negative ion mode. After processing the raw data for quality control and database annotation, a total of 303 metabolites were detected in the positive ion mode, 286 metabolites in the negative ion mode, and a total of 589 metabolites in both ion modes.

PLS-DA indicated distinct metabolite differences among the ZD and CK groups (Figure 4A). Furthermore, Figure 4B depicts the contrasting metabolites between the two groups. In the ZD group, the levels of Alline, Saccharin, 4-Hydroxybenzenesulfonic Acid, Prilocaine, Cannabidiolic Acid, Naringin, Tomatidine, Glycodeoxycholic Acid, Taurocholic Acid, Palmitoleoyl Ethanolamide, 4-Coumaric Acid, delta (17)-6-Ketoprostaglandin F1alpha, Ursocholanic Acid, N6,N6,N6-Trimethyl-L-lysine, and Lignoceric Acid were significantly higher compared to the control group (*p* < 0.05). In the CK group, the levels of *N*-Acetylsphingosine, Luteolin, 3-hydroxy-C10-homoserine Lactone, 4-hydroxyphenyl heptane-3,5-diol, 3-oxo-C12 Homoserine Lactone, Isorhamnetin, Corymbosin, 2-Methoxyestrone, All Trans Retinal, Ascorbic294 Acid, Glycitein, Crotetamide, 6-Methylquinoline, and Vitexin were significantly higher (*p* < 0.05). Figure 4C shows the KEGG pathways that were enriched in differential metabolites in the ZD group in contrast to the CK group. Colorectal cancer pathways were enriched in the ZD group (*p* < 0.05). Recurrent clostridium difficile infection, Cranioclavicular dysplasia syndrome, Lleal Crohn’s disease, Metastatic melanoma, clostridium difficile infection, Crohn’s disease, and Ulcerative colitis were higher in the ZD group, though the differences were not statistically significant.

## 4. Discussion

Micronutrient deficiencies, also known as “hidden hunger”, impacts approximately 2 billion people, with deficiencies of iron, zinc, folic acid, and vitamin A being the main cause of the problem [36]. Among them, zinc was the second-most essential micronutrient in the human body after iron [37]. It was reported that zinc deficiency influenced 17% of the global human population, and about one billion people suffered from zinc deficiency in China [6,38]. In school-age children at high risk of being zinc-deficient, whose nutritional status has always been the focus of social attention, especially in rural areas where the economy is underdeveloped, the prevalence of zinc deficiency was even higher due to insufficient medical standards and a shortage of nutritional knowledge [39]. Earlier studies have showed the impacts of micronutrients on the gut flora in different models [40] and suggested that unbalanced micronutrients may result in the disruption of the composition and structure of the gut microbiome as well as the bloom of *Enteropathogens* and the alteration of bacterial metabolites. Therefore, the microbiome and metabolome may be useful biomarkers for assessing and predicting the status of micronutrients. This study investigated the profiles of the gut flora and metabolites under zinc deficiency using gut metagenomics and metabolomics, and it explored the potential effect of certain dominant bacterial species and metabolites on the health of zinc-deficient children.

Currently, there are limited investigations into the effect of zinc inadequacy on intestinal microbiota. In the chick experiment, the zinc-supplemented group had a higher relative abundance of *Firmicutes*, while *Bacteroidetes* and *Proteobacteria* were more abundant in the zinc-deficient group [15]. Furthermore, within the intestinal tract of zinc-deficient mice, the relative abundance of *Actinobacteria* and *Proteobacteria* was significantly higher, and the relative abundance of *Firmicutes* was significantly lower. However, the relative abundance of microorganisms such as *Bacteroidetes* and *Verrucomicrobia* was not significant between the two groups [18]. In another experiment in pregnant mice, a zinc-deficient diet resulted in an increased relative abundance of *Actinobacteriahe* and *Bacteroidetes*. In addition, the relative abundance of *Firmicutes* was significantly increased in both the zinc-deficient and zinc-suppressed diets compared with the control group, while the relative abundance of *Verrucomicrobia* was reduced in the zinc-deficient diet group and increased in a significant way in the zinc-suppressed diet group [16]. Consistent with the results of the above studies, we found that the relative abundances of *Bacteroidetes* and *Actinobacteriota* were significantly higher in the ZD group, which may be due to the fact that zinc-deficient environments lead to a reduced diversity of the flora, which is preferentially made up of bacterial species that survive under zinc-limited conditions. In contrast to the results of all the above studies, we found no difference in the relative abundance of *Firmicutes* between the two groups. This study indicated that specific bacterial groups, including *Firmicutes*, could survive in a low-zinc setting by decreasing their zinc demand and employing a more advantageous zinc absorption process, which could explain the notable rise in *Firmicutes* observed in both the zinc-deficient and zinc-suppressed diet cohorts [16]. By contrast, the increased relative abundance of *Firmicutes* in the zinc-supplemented group may be a result of differences in experimental subjects, which in the available studies included chickens, mice, and so on. During a four-week zinc dietary intervention experiment, it was found that high and excess zinc diets led to an increase in the relative abundance of *Verrucomicrobia*, with the proportion of *Proteobacteria* being notably higher in the low- and high-zinc-diet groups compared to the control and high-zinc-diet groups. During an eight-week zinc diet intervention experiment, mice with the high-zinc-diet exhibited a high abundance of *Actinobacteria*. Conversely, the low-zinc-diet group had a significantly increased proportion of *Verrucomicrobia*, while the high-zinc-diet group experienced a significant decrease in their proportion [13]. Additionally, the study determined that *Desulfovibrio* sp. strain ABHU2SB demonstrated a significant association with serum zinc levels during both short- and long-term interventions, thereby indicating its potential as a biomarker for zinc status. When the above studies on zinc and gut microbiota are compared, the results are mixed and some are even completely opposite. The possible causes for these variances could be attributed to the following: (1) Differences in the zinc levels of the intervention diets. The diets encompassed zinc-deficient, zinc-inhibitor, low-zinc, high-zinc, and excessive-zinc diets, amongst others. (2) Differences in the duration of the intervention. The duration of zinc diets in previous studies has ranged from three to eight weeks.

In the present research, various intestinal microbes identified in the ZD and CK groups also play an essential role in human health and disease. Among them, intestinal microorganisms such as *Phocaeicola vulgatus*, *Pseudomonas putrefaciens*, *Bacteroides uniformis*, *E.* regularis, *E. przewalskiii*, and *E. ruminalis* contained genes that encode ABC transporter [41,42], and the transmembrane transport of zinc ions was associated with the ZnuABC transporter [43]. Other studies have found that certain bacteria can produce ZnuABC transporter proteins in low-zinc conditions [44], thus giving them an advantage in competing for zinc ions through these proteins. In addition, certain gut microorganisms, such as *Phocaeicola vulgatus*, *Pseudomonas putrefaciens*, *Bacteroides uniformis*, *Phocaeicola* sp000434735, *Prevotella*, and *Ruminalococcus*, contained genes that encode glycoside hydrolases (GHs). The enzymes could degrade glycosylated mucins within the intestinal mucus layer [45]. The accumulation of a large number of the above gut flora was highly likely to result in the deterioration in the gut mucus layer and harm to the gut barrier. Among the dominant microorganisms in the CK group, *Blautia* comprised anaerobic bacteria with probiotic properties that were abundant in the mammalian gut and feces. Some of these characteristics have been observed in current research, such as the ability of *Blautia* to impede the transmission and infection of drug-resistant pathogens, and the production of short-chain fatty acids, which are essential for maintaining intestinal balance and preventing inflammation [46]. In addition, various studies have demonstrated that an elevated relative abundance of *Blautia* decreases levels of inflammatory cytokines and reduces body mass index [47,48,49], which is consistent with the increased secretion of the inflammatory factor TNF-α in the ZD group in the present study. *Bifidobacteria* were among the first microorganisms to colonize the human gut naturally and had beneficial effects on the digestive, immune, and nervous systems, rendering them closely linked with health [50]. *Bifidobacteria* could produce B vitamins, including vitamin B1, vitamin B2, and vitamin B12, among others [51]. Additionally, they could alleviate lactose intolerance [52], prevent acute diarrhea and colorectal cancer [53], reduce serum cholesterol [54], and help prevent and treat inflammatory bowel disease [55]. In addition, *Bifidobacteria* and *Lactobacillus*, identified as probiotics, have demonstrated an active role in infections and inflammatory bowel disease. They were capable of improving inflammatory responses, reducing TNF-α levels, and preventing increased intestinal epithelial permeability and epithelial cell apoptosis [56,57]. *Coprococcus* can assess human gastrointestinal health as a biomarker [58], and it has also now been shown that *Coprococcus eutactus*, a promising probiotic, can ameliorate colitis by reducing the levels of the pro-inflammatory cytokines TNF-α and IL-1β, and increasing the levels of the anti-inflammatory factors IL-5 and IL-10. Furthermore, *C. eutactus* also protects the gut mucosa and epithelial barrier by increasing levels of goblet cells, mucin families, and tight junction proteins [59]. *Bacteroides uniformis* and *Anaerobutyricum hallii* have shown therapeutic potential as next-generation probiotics (NGPs) for the treatment of metabolic disorders, obesity, and obesity-related diseases [60,61]. Possible mechanisms for improvement include modulating intestinal microbiota and intestinal peptide secretion, improving gut microbiota dysbiosis and gut barrier function, and reducing chronic low-grade inflammation [60].

In this study, the untargeted LC-MS findings demonstrated distinct metabolomic profiles within the ZD and CK groups. Saccharin is a widely used artificial sweetener in foods, including carbonated beverages, fruit juices, candies, and biscuits [62], and it may, therefore, be consumed more frequently by children than adults. The overconsumption of saccharin could result in various health issues, including metabolic disorders and glucose intolerance [63]. In mice models, saccharin may disrupt the gut microbiota ecology, impacting their metabolic roles and promoting inflammation [64]. Also, it may decrease the amount of tight junction proteins, contributing to the increase in gut permeability [65]. As a primary 12α-hydroxylated bile acid, taurocholic acid could induce permeability in the distal intestine of rats, mechanistically increase phosphorylation of myosin light chain 2, and reduce the expression of claudin genes in the ileal epithelium [66]. All-trans retinol, commonly referred to as vitamin A, plays a crucial role in human visual function by contributing to the preservation of photoreceptors that aid in dark vision and circulate within the visual cells. Zinc is one of the most abundant micronutrients in human ocular tissues, and it is distributed in tissues including the retina, choroid, ciliary body, iris, and optic nerve [67]. Zinc is crucial to enzyme function in ocular tissues, specifically in retinaldehyde reductase, a zinc-binding dehydrogenase found in the retina. Animal studies indicated that severe zinc deficiency could heavily impact the function of the enzyme [68]. Zinc deficiency may also reduce retinol-binding protein concentration in the plasma and liver, which could hinder the body’s ability to mobilize vitamin A from hepatic stores, which may have a negative impact on vitamin A metabolism at the cellular level [68]. Ascorbic acid, commonly referred to as vitamin C, is a crucial micronutrient that participates in metabolic reactions within the body. Vitamin C could scavenge reactive oxygen radicals produced during inflammation due to its potent antioxidant function, and it also plays a pivotal role in reducing apoptosis during cellular senescence. Additionally, vitamin C could regulate cell signaling pathways and has anti-tumor properties [69]. It has also been shown that ascorbic acid treatment suppresses the expression of inflammatory markers (sepsis and TNF-α) and markers of cellular damage, which is in accordance with the decreased levels of TNF-α in the CK group [70].

In this study, the KEGG pathways enriched for specific metabolites in the ZD group were different from that of the CK group and may be associated with many intestinal diseases. Among them, colorectal cancer (CRC) is one of the malignant tumors with the highest morbidity and mortality rates. Dietary, genetic, and pathological factors are thought to contribute to the development of CRC, although the exact mechanism has not yet been determined. A population-based study has shown that serum zinc was less abundant in CRC patients than in healthy individuals [71]. One study has observed a reduction in plasma zinc and the activity of zinc-containing enzymes in rats with colon cancer. The presence of pre-cancerous lesions in the colon was associated with zinc-related enzymes, and the above indicators will gradually decline as the cancer progresses [72]. Moreover, it was reported that zinc intake affected the prevalence of CRC. A study showed a significant association of zinc consumption and the risk of CRC in female patients, although no significant correlation was observed in male patients. Furthermore, research has shown a U-shaped correlation between the amount of zinc consumption and the vulnerability to CRC [73]. Inflammatory bowel disease (IBD) includes Ulcerative colitis (UC) and Crohn’s disease (CD), where the cause is unknown. Studies have indicated that patients diagnosed with CD and UC tend to have reduced zinc intake and serum zinc concentrations [74,75,76]. Adverse disease-specific outcomes were more common in patients with CD and UC who had serum zinc deficiency, which could be improved after the normalization of zinc levels [76]. However, a study discovered low levels of zinc in the serum which were correlated with CD but unrelated to UC [77]. Likewise, it was found that dietary zinc supplements had a negative correlation with CD risk, but not with UC [78]. Research indicated that zinc deficiency may lead to multiple recurrent *Clostridioides difficile* infections in patients, whereas zinc supplementation improved symptoms and reduced *C. difficile* recurrence [79]. A study examined the correlation of zinc intake levels and *C*. *difficile* recurrence after FMT treatment. The *C. difficile* infection rate was 8% in the low-zinc group after zinc supplementation, whereas patients without supplementation had a rate of 50%. This study indicated that zinc deficiency is a crucial factor in causing *C. difficile* to recur [80].

## 5. Conclusions

In this study, microbiomics and metabolomics were conducted to examine the effect of zinc deficiency on the intestinal microflora and metabolites in school-age children. The result revealed that there were significant differences in the composition and function of intestinal microorganisms and metabolite levels between ZD and CK groups, suggesting that zinc deficiency may interfere with the intestinal flora of children, affecting its metabolic function, and resulting in altered levels of intestinal metabolites. Furthermore, *Phocaeicola vulgatus*, *Bacteroidetes uniformis*, *Roseburia intestinalis*, and other mucin-degrading flora were more abundant in zinc-deficient children, and higher levels of taurocholic acid and saccharin associated with intestinal barrier damage may ultimately lead to intestinal barrier damage and an intestinal pro-inflammatory response, subsequently leading to the malabsorption of nutrients and further exacerbating the zinc deficiency in the body.

There are also some limitations to this study. First, small samples may influence the generalizability of the findings. Second, as an observational study, this study could provide clues for further etiological studies, but was limited in determining causality. Animal models such as mice based on population studies could be used to validate the causal link between gut flora and host well-being in the future, to delve into the action mechanism through zinc-deficient mouse models, and to determine potential gut microbe and metabolite biomarkers related to zinc deficiency.

## Figures and Tables

**Figure 1 nutrients-16-01289-f001:**
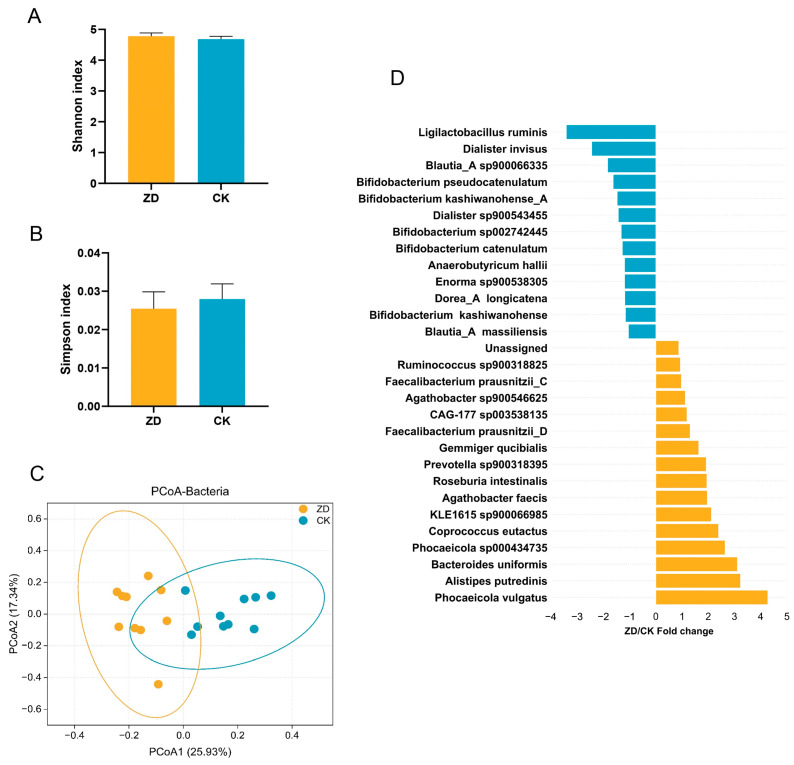
Comparisons of alpha diversity measured by (**A**) Shannon index and (**B**) Simpson index and beta diversity using (**C**) PCoA based on Bray–Curtis distance. (**D**) The significantly different microbial taxa (*p* < 0.05) in the relative abundance between ZD and CK groups. ZD, zinc deficiency; CK, control; PCoA, Principal Coordinate Analysis.

**Figure 2 nutrients-16-01289-f002:**
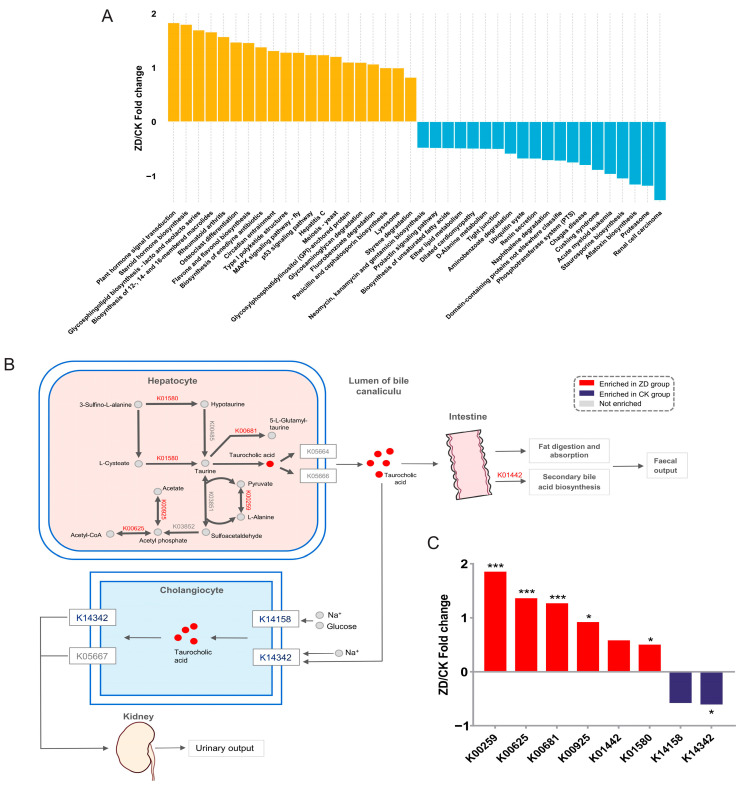
Differential KEGG functions between ZD and CK groups. (**A**) ZD/CK fold change of significantly enriched microbial KEGG pathways (*p* < 0.05). The top 20 enriched pathways in both groups are presented. (**B**) Illustration of taurine and hypotaurine metabolism in ZD and CK groups and (**C**) the relative changes in the corresponding KOs. Red and blue indicate significant alterations selectively in ZD and CK, respectively. * *p* < 0.05 and *** *p* < 0.001.

**Figure 3 nutrients-16-01289-f003:**
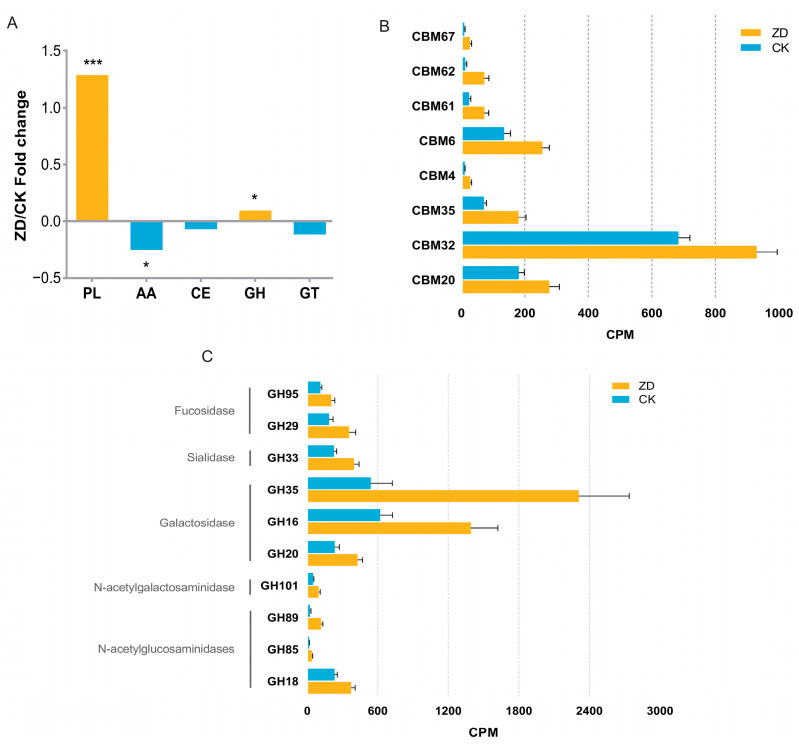
The carbohydrate-active enzymes of ZD and CK groups. (**A**) The abundance comparison of GH (glycoside hydrolase), GT (glycosyl transferase), CE (carbohydrate esterases), PL (polysaccharide lyases), and AA (auxiliary activities). * *p* < 0.05 and *** *p* < 0.001. (**B**) The differential CBM (carbohydrate-binding module) between ZD and CK subjects (*p* < 0.05). (**C**) The significantly different GH family in two groups (*p* < 0.05).

**Figure 4 nutrients-16-01289-f004:**
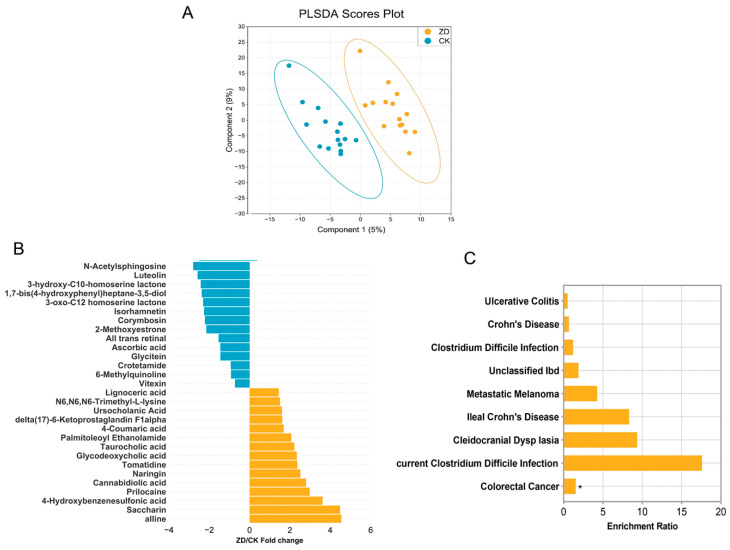
Fecal metabolome of ZD and CK groups. (**A**) Partial least-squares discriminant analysis (PLS-DA) revealed the profile of the metabolome in the ZD and CK groups. (**B**) The abundance of differential fecal metabolites in the two groups is presented by the ZD/CK fold change according to DESeq2 analysis (*p* < 0.05). (**C**) The KEGG pathways that were enriched from ZD-specific metabolites as compared with CK subjects. * *p* < 0.05.

**Table 1 nutrients-16-01289-t001:** Characteristics of zinc-deficient and control children (n= 30).

	ZD (n = 15)	CK (n = 15)	*p*
Age, years (mean ± SEM)	8.20 ± 0.28	9.07 ± 0.40	0.085
Sex (females, males)	8 females, 7 males	7 females, 8 males	1.000
Z-score			
Median HAZ score	0.20	−0.30	0.221
Median WAZ score	0.02	−0.48	0.136
Median BMIZ score	−0.06	−0.62	0.059
Blood indices (mean ± SEM)			
Serum zinc (μg/dL)	22.94 ± 4.67	150.59 ± 7.49	<0.001
IL-6 (pg/mL)	18.29 ± 1.72	18.07 ± 2.86	0.949
TNF-α (pg/mL)	13.86 ± 0.73	7.44 ± 1.14	<0.001
IL-1β (pg/mL)	20.26 ± 3.88	24.06 ± 4.97	0.551

*p* < 0.001.

## Data Availability

The raw sequence reads were uploaded to the European Nucleotide Archive (https://www.ebi.ac.uk/ena/browser/home, accessed on 22 December 2023) under the project number PRJEB71009. The original data of the metabolic group have been uploaded to the Metabolights Database (https://www.ebi.ac.uk/metabolights/, accessed on 5 January 2024), and the project number is MTBLS9247, URL: http://www.ebi.ac.uk/metabolights/MTBLS9247, accessed on 5 January 2024.

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
