# Peer review of "Intestinal Barrier Impairment Induced by Gut Microbiome and Its Metabolites in School-Age Children with Zinc Deficiency"

_nutrients, 2024, doi:10.3390/nu16091289_

Round 1

Reviewer 1 Report

Comments and Suggestions for Authors

the study provides insights into the impact of zinc deficiency on intestinal microflora and metabolites in school-age children. However, additional research is needed to address the limitations and further elucidate the mechanistic links between zinc status, gut microbiota composition, and intestinal health.

It is important to take into account several limitations:

Establishing a causal relationship between zinc deficiency and changes in the gut microbiome and metabolites is hampered by the cross-sectional nature of the study. The temporal link between zinc status and gut health outcomes in school-age children would be better understood through longitudinal investigations.

Additional information on other relevant factors, such as socioeconomic status, dietary habits, and health status, would provide a more comprehensive understanding of the study population and potential confounding variables.

When interpreting the study's findings, a number of limitations should be taken into account. The generalizability of the findings has been put into question by the inadequate description of the study population's demographics and sample size. Furthermore, the study's main focus is observational data, which makes it difficult to determine a causal association between zinc deficiency and the overall composition of the gut microbiota.

Author Response

General comments: the study provides insights into the impact of zinc deficiency on intestinal microflora and metabolites in school-age children. However, additional research is needed to address the limitations and further elucidate the mechanistic links between zinc status, gut microbiota composition, and intestinal health.

It is important to take into account several limitations:

Response: We are sincerely grateful to reviewer 1 for thoroughly reviewing our manuscript. We have carefully considered the suggestions of reviewer and made some changes in the revised manuscript. All the changes in the revised manuscript are highlighted in yellow. Please find the following detailed responses to your comments.

Point 1:

Establishing a causal relationship between zinc deficiency and changes in the gut microbiome and metabolites is hampered by the cross-sectional nature of the study. The temporal link between zinc status and gut health outcomes in school-age children would be better understood through longitudinal investigations.

Response: Thank you for your comments. The research was mainly carried out in 2020, and due to the COVID-19 pandemic at that time, we were unable to conduct longitudinal study. The causal relationship between zinc deficiency and changes in the gut microbiome and metabolites cannot be clarified by cross-sectional study, but it could provide clues for further studies. Actually, we are preparing our next manuscript which is mainly focused on animal model, which explore the relationship between zinc deficiency and gut flora and metabolites. In mouse experiments, we set up two mouse models, a pseudo-germfree mouse model and a faecal microbiota transplantation mouse model, to elucidate the mechanism of action of zinc deficiency and dietary supplementation with ZnAAs by affecting intestinal microecology and thus causing pathological changes. The results showed that the zinc deficiency and zinc adequacy groups of both models were significantly separated with different gut flora structures. Among them, the abundance of gut microorganisms negatively associated with depression, such as Eubacterium, Clostridia, Clostridiales, Coprococcus, and Dorea, was significantly lower in the zinc deficiency group, and the relative abundance of Oscillospira, a microorganism implicated in Fragile X syndrome, one of the major triggers of autism, was significantly lower in zinc deficiency group. In addition, 13KEGG metabolic pathways, including African trypanosomiasis, Staphylococcus aureus infection, Ubiquinone and other terpenoid-quinone biosynthesis, were significantly regulated in the fecal microbiota transplantation zinc deficiency group; D-arginine and D-ornithine metabolism, Flavonoid biosynthesis, Steroid biosynthesis and Amoebiasis were significantly down-regulated in zinc deficiency group of fecal microbiota transplantation. The KEGG pathway was significantly up-regulated only for Other types of O-glycan biosynthesis in pseudo-germfree zinc deficiency group.

Point 2:

Additional information on other relevant factors, such as socioeconomic status, dietary habits and health status, would provide a more comprehensive understanding of the study population and potential confounding variables.

Response: Thank you very much for your constructive suggestions. More information about the subjects in this study can be found in another published article [1]. In previous studies, we screened children for their health status to ensure as far as possible that any factors that might interfere with the results were excluded. Criteria used to select: (1) No history of acute or chronic illness (cancer, malignant tumors, kidney disease, heart disease, diabetes, or liver disease, etc). (2) No gastrointestinal symptoms (constipation, diarrhea, etc) in the past three months and no history of chronic gastrointestinal diseases (gastrointestinal tumor, etc). (3) No other chronic diseases. (4) No any infectious diseases (AIDS, hepatitis b, hepatitis c, and syphilis, etc). (5) No history of intake of antibiotics, antiviral, antifungal or analgesic drugs in the past three months. (6) No fever symptoms in the past three months. (7) No nutritional supplements (zinc, calcium, vitamin, etc) have been taken in the past three months. 41 control children and 26 zinc-deficient children were included.

For the most influencing factors of intestinal microbiota, such as age, gender, dietary intake level, etc., difference analysis was conducted between the two groups, and the previous study [1] showed that there was no significant difference between the two groups, which have minimized the effects of confounders. From the 67 children reported in the previous study [1], 30 children were selected for intestinal microbiome and metabolites in the present study.

Reference:

[1] Chen X, Jiang Y, Wang Z, et al. Alteration in Gut Microbiota Associated with Zinc Deficiency in School-Age Children [J]. Nutrients, 2022, 14(14).

Point 3:

When interpreting the study's findings, a number of limitations should be taken into account. The generalizability of the findings has been put into question by the inadequate description of the study population's demographics and sample size. Furthermore, the study's main focus is observational data, which makes it difficult to determine a causal association between zinc deficiency and the overall composition of the gut microbiota.

Response: Thank you for your comments. In recent years, we have carried out a series of studies about zinc deficiency and gut microbiome and metabolites. In the first published manuscript, 16S rDNA gene sequencing was conducted to explore the relationship between zinc deficiency and the gut microbiome in school-age children. In the second manuscript, which was the present study, we took further investigation through metagenome and metabolome. And the third manuscript we are preparing is mainly focused on mouse model and sterile mouse model transplanted with human gut microbiota, which systematically explored the relationship between zinc deficiency and gut flora and metabolites. Through the above series of related studies, we hope to clarify the effects ofzinc deficiency on the gut microbiome and metabolites, and to reveal the relationship between the two. In our conclusions, we clearly stated the limitations of this study, which were highlighted in yellow.

Reviewer 2 Report

Comments and Suggestions for Authors

In this paper, the authors investigated the effect of zinc deficiency on the gut microbiome and its metabolites. This work is significant if the following issues can be addressed:

1. Figure 1, add 95% confidence ellipse in the PCoA plot;

2. Figure 2, specify the significance results are among which groups.

3. Figure 3, statistical analysis in panel B and panel C is missing.

4. Similar opinions go for Figure 4.

Uniform all the cited references. some references don't have page number, some references don't have issue number.

Author Response

General comments:

In this paper, the authors investigated the effect of zinc deficiency on the gut microbiome and its metabolites. This work is significant if the following issues can be addressed:

Response: We are very honored that reviewer 2 endorsed our manuscript. And sincere thanks should be given to you for the constructive comments and suggestions. We have carefully considered the suggestions of reviewer and made some changes in the revised manuscript. All the changes in the revised manuscript are highlighted in yellow. We have provided a point-by-point response to the reviewers' comments below in red color.

Point 1:

Figure 1, add 95% confidence ellipse in the PCoA plot;

Response: Thanks a lot for your suggestion. We have added 95% confidence ellipse in the PCoA plot. Please refer to Figure 1.

Point 2:

Figure 2, specify the significance results are among which groups.

Response: Thank you very much for your comments. In Figure 2(A), the presented top20 pathways were all significantly different between ZD and CK group. In Figure 2(C), the KO with an asterisk were significantly different between ZD and CK group.

Point 3:

Figure 3, statistical analysis in panel B and panel C is missing.

Response: Thank you for your suggestion. We have added statistical analysis on lines 123~127 and 138~140 in the manuscript, which were highlighted in yellow.

Point 4:

Similar opinions go for Figure 4.

Response: Thanks a lot for your suggestion. We have added 95% confidence ellipse in Figure 4(A).

Point 5:

Uniform all the cited references. some references don't have page number, some references don't have issue number.

Response: Thank you for your comments. We have checked all the cited references and added missing page number or issue number.

Round 2

Reviewer 1 Report

Comments and Suggestions for Authors

The authors have provided concise and convincing arguments addressing the concerns raised, demonstrating their thorough understanding of the subject matter. Additionally, their expertise and contributions to the field are evident in their responses and the manuscript itself.

Therefore, considering the authors' diligent efforts and the significance of their work in the field, I believe that the manuscript is ready for publication. I would like to extend my gratitude to the authors for their dedication and diligence throughout the review process.